# Cytokine Profile and Anti-Inflammatory Activity of a Standardized Conditioned Medium Obtained by Coculture of Monocytes and Mesenchymal Stromal Cells (PRS CK STORM)

**DOI:** 10.3390/biom12040534

**Published:** 2022-03-31

**Authors:** Juan Pedro Lapuente, Alejandro Blázquez-Martínez, Joaquín Marco-Brualla, Gonzalo Gómez, Paula Desportes, Jara Sanz, Pablo Fernández, Mario García-Gil, Fernando Bermejo, Juan V. San Martín, Alicia Algaba, Juan Carlos De Gregorio, Daniel Lapuente, Almudena De Gregorio, Belén Lapuente, María de la Viñas Andrés, Alberto Anel

**Affiliations:** 1R4T Molecular and Cell Biology Research Laboratories, Fuenlabrada Hospital, 28942 Madrid, Spain; alejanb@gmail.com (A.B.-M.); gonzalog628@gmail.com (G.G.); pablo.fv@outlook.com (P.F.); jcdegregorio@cres.com.es (J.C.D.G.); daniel_lapuente_hernandez@yahoo.es (D.L.); almudenadegregorio@gmail.com (A.D.G.); belelapu@gmail.com (B.L.); v.andres@livingcells.org (M.d.l.V.A.); 2Group Immunity, Cancer and Stem Cells, Faculty of Sciences, University of Zaragoza, 50009 Zaragoza, Spain; joaquin_marco_91@hotmail.com; 3GMP Facility, Peaches Biotech, 28050 Madrid, Spain; paula_phisiup@hotmail.com (P.D.); jara_phisiup@hotmail.com (J.S.); 4Pharmacy Department, Fuenlabrada Hospital, 28942 Madrid, Spain; mgarciagil@salud.madrid.org; 5Digestive Department, Fuenlabrada Hospital, 28942 Madrid, Spain; fernando.bermejo@salud.madrid.org; 6Medicine Department, University Rey Juan Carlos, Fuenlabrada Hospital, 28942 Madrid, Spain; 7Internal Medicine Department, Fuenlabrada Hospital, 28942 Madrid, Spain; juanvictor.san@salud.madrid.org; 8Clinical Assay Department, Fuelabrada Hospital, 28942 Madrid, Spain; alicia_algaba@hotmail.com

**Keywords:** MSC, monocyte, coculture, crosstalk, secretome, inflammation

## Abstract

Intercellular communication between monocytes/macrophages and cells involved in tissue regeneration, such as mesenchymal stromal cells (MSCs) and primary tissue cells, is essential for tissue regeneration and recovery of homeostasis. Typically, in the final phase of the inflammation-resolving process, this intercellular communication drives an anti-inflammatory immunomodulatory response. To obtain a safe and effective treatment to counteract the cytokine storm associated with a disproportionate immune response to severe infections, including that associated with COVID-19, by means of naturally balanced immunomodulation, our group has standardized the production under GMP-like conditions of a secretome by coculture of macrophages and MSCs. To characterize this proteome, we determined the expression of molecules related to cellular immune response and tissue regeneration, as well as its possible toxicity and anti-inflammatory potency. The results show a specific molecular pattern of interaction between the two cell types studied, with an anti-inflammatory and regenerative profile. In addition, the secretome is not toxic by itself on human PBMC or on THP-1 monocytes and prevents lipopolysaccharide (LPS)-induced growth effects on those cell types. Finally, PRS CK STORM prevents LPS-induced TNF-A and IL-1Β secretion from PBMC and from THP-1 cells at the same level as hydrocortisone, demonstrating its anti-inflammatory potency.

## 1. Introduction

Intercellular communication between macrophages and cells involved in tissue regeneration, such as mesenchymal stromal cells (MSCs) and primary tissue cells, is essential for tissue regeneration and recovery of homeostasis in the extracellular matrix [1,2]. When tissue injury occurs, monocytes are recruited from the peripheral circulation, penetrating injured tissues, where they differentiate into macrophages [3]. Macrophages respond to paracrine signals in the resident tissue and can acquire either a proinflammatory (M1 or classical activation) or anti-inflammatory (M2 or alternative activation) phenotype [4]. These M1 or M2 phenotypes will determine the polarized responses of other lymphocytes through toll-like receptor (TLR) signaling [5]. M2 macrophages, which can generate immunotolerance and anti-inflammatory and antifibrotic effects [6,7] in different tissues, are able through their secretome to activate regulatory T cells (Tregs), a subset of T cells essential for enhancing immune tolerance [8]. In general, M2 macrophages play an anti-inflammatory and immunomodulatory role [7], and their activation can be induced by multiple stimuli, usually through their pattern-recognition receptors (PRRs). These stimuli can come from external pathogens and recognize molecular structures from viruses, bacteria, fungi and other pathogens, known as pathogen-associated molecular patterns (PAMPS) [9,10,11], and from self-produced molecular structures derived from damage to tissues and cells, known as damage-associated molecular patterns (DAMPS) [12,13,14,15,16]. In summary, M2 macrophages are characterized by a high phagocytic capacity, secretion of extracellular matrix (ECM) components, angiogenic and chemotactic factors [17], and anti-inflammatory and immunomodulatory cytokines [18,19,20]. All these factors promote tissue regeneration and homeostasis.

In the resolution phase of inflammation, in which tissue damage is eliminated and tissue integrity and functionality are restored, maintenance of the M2 phenotype is essential—this is supported by different cells, such as eosinophils, tissue-resident innate lymphoid cell type 2 (ILC2), and MSCs [21,22,23]. MSCs are involved in the maintenance and/or restoration of tissue homeostasis, always working in coordination with the immune system, mostly macrophages [24,25]. The stimuli of certain cytokines and chemokines secreted, among others, by macrophages, will trigger the coordinated reaction of MSCs that will produce a plethora of growth factors, chemokines, and cytokines capable of recruiting and regulating the participation of endothelial cells, fibroblasts, and other progenitors to promote tissue regeneration, exerting an immunomodulatory function on inflammation [26].

When MSCs are exposed to macrophages (monocytes attached to a substrate) in a coculture without direct contact, they cause macrophage polarization to the M2 type, providing a model for studying regenerative conditions in vivo [27,28]. On the other hand, the immunomodulatory influence of MSCs on M2 macrophages has been demonstrated. For example, when MSCs are incubated with IL-1β, COX-2 expression is increased, suggesting that there is feedback between IL-1β in the inflammasome and the immunosuppressive capacity of MSCs, with direct COX-2/PGE2 signaling being responsible for the immunosuppressive effects on the inflammasome [29]. It has also been shown that activation of the NF-kβ pathway in macrophages stimulated with bacterial lipopolysaccharide (LPS) is repressed by coculture with MSCs [30]. This response is mediated by a decrease in the relative expression of the IKK kinases implicated in TLR4-mediated activation of the NF-kβ pathway [30]. Another study showed that the expression of anti-inflammatory surface markers such as B220 and CD206, typical of M2 macrophages, was increased in MSC-stimulated macrophages [31,32].

In the late 1970s, the ability of infectious agents (bacteria or viruses) to trigger cytokine storm syndrome was first described with the recognition of a series of cases of hemophagocytic lymphohistiocytosis of viral origin [33]. This cytokine storm is basically characterized by an exaggerated production of soluble proinflammatory and profibrotic mediators (especially IL-1β, IL-6, and TNF-α) together with an aberrant immunopathological reaction involving an incoordination between the innate and adaptive immune systems, with an overactivation of the innate immune system [34,35,36,37,38]. Because of this cytokine storm, a situation of multi-organ hyperinflammation is triggered, usually mainly affecting the lung and pancreas, among other organs, and often leading to acute respiratory distress syndrome and/or acute lung injury, which can result in multi-organ failure.

Despite the known association between high levels of proinflammatory and profibrotic cytokines and chemokines with high morbidity and mortality rates after an infectious process, there is no suitable drug to treat the cytokine storm [38]. In addition, adverse effects associated with chimeric antigen receptor (CAR)-T treatment of leukemia have been also classified as a cytokine storm disease [39]. Recently, the COVID-19 pandemic has been characterized in its late and mortal phase III as a hyper-inflammatory disease [40]. To overcome this important health problem, the proposal advocated by our group for the prevention and control of infection-associated cytokine storms, including that associated with COVID-19, is the therapeutic use of a secretome derived from the coculture of M2 macrophages with MSCs. This secretome offers a naturally balanced composition of growth factors, cytokines, and chemokines with anti-inflammatory activity respecting natural pleiotropic relationships.

## 2. Materials and Methods

### 2.1. Primary Cell Isolation, Expansion, and Coculture

#### 2.1.1. MSCs Isolation

A lipoaspirate sample was obtained from a 38-year-old, healthy female donor who underwent elective liposuction after obtaining informed consent (standard, following the directives of Royal Decree-Law 9/2014 of 4 July, establishing quality and safety standards for the donation, procurement, testing, processing, preservation, storage, and distribution of human tissues and cells). In addition, SARS-CoV2 infection was ruled out by PCR analysis of a nasopharyngeal sample taken 24 h prior to the procedure. Each lipoaspirate sample was processed according to the established method [41,42] to obtain the vascular-stromal fraction (SVF). The lipoaspirate sample was washed twice with phosphate buffered saline (PBS, Gibco-BRL, Grand Island, NY, USA), centrifuged at 250× *g* for 10 min to eliminate anesthetic compounds, and adipose tissue was digested with 0.075% collagenase type I (Gibco-BRL, Grand Island, NY, USA) for 30 min at 37 °C and 250 rpm agitation. After digestion, the sample was centrifuged at 250× *g* for 10 min and the cell pellet was retrieved. Remaining erythrocytes were eliminated by incubating the pellet with lysis buffer (160 mM NH_4_Cl, 10 mM KHCO_3_ and 1 mM EDTA; all from Sigma-Aldrich, St Louis, MS, USA) for 10 min at room temperature. After lysis, cells (SVF) were centrifuged at 250× *g* for 10 min, resuspended and seeded in cell-culture treated flasks at 3 × 10^4^ cells/cm^2^ in complete culture medium (DMEM non-essential amino acids + 10% FBS + 1% P/S) and kept at 37 °C, 5% CO_2,_ and 98% RH. After 24 h the culture surface was washed with tempered PBS to remove nonadherent cells, and the adherent cell population was cultured to sub-confluence under the same conditions as above, changing the culture medium 3 times a week. Trypsin 0.05% (Gibco) was used to lift the cells and proliferation medium (DMEM+ 1% P/S + 2.5% platelet lysate (Sigma-Aldrich)) was used for subculture, making the necessary passages until a homogeneous population of MSCs was obtained (typically at passage 3 so that when the cells are thawed, they remain at passage 4). After culture, cells were frozen on a freezing ramp of −0.5 °C/minute to −80 °C in freezing medium consisting of 10% dimethyl sulfoxide (DMSO, Sigma) in FBS, then immersed in liquid N_2_ for future use. In our research, MSC cells with 4 culture passages were used in all cases.

#### 2.1.2. PBMC and Monocyte Isolation

For the isolation of peripheral blood mononuclear cells (PBMCs), a density gradient centrifugation was performed with Ficoll–Hystopaque 1077 Premium (GE Healthcare, Wauwatosa, WI, USA) using a buffy coat from a donation from the Castilla La Mancha Blood Bank. Buffy coats were diluted 1:1 with physiological saline and layered on top of the gradient at a 4:3 buffy coat to density gradient ratio. Tubes were centrifuged at 400× *g* for 30 min at RT without using the centrifuge break. After centrifugation, the PBMC layer was collected in clean tubes, washed with saline, and contaminating erythrocytes were removed with lysis buffer, each wash step was carried out by centrifugation at 250× *g* for 10 min. The obtained PBMCs were either used for in vitro assays or resuspended in monocyte culture medium (CTS-AIM-V™, Gibco-BRL, Waltham, MA, USA) by seeding these at 2.5 × 10^6^ cells/mL in cell culture flasks and incubating them for 1.5 h at 37 °C, 5% CO_2_ and 98% RH. Once adherent, cells were gently washed with tempered saline to eliminate nonadherent cells, and remaining cells were lifted with a cell scraper (Corning, Corning, NY, USA). Collected cells were seeded onto Transwell^®^ inserts (Falcon, PET, 1 μM pore size) at a density of 500,000 cells/cm^2^ and cultured for 4 days in monocyte culture medium supplemented with 10ng/mL M-CSF (R&D Systems, Minneapolis, MN, USA). After 4 days, the culture medium was discarded, inserts washed with tempered PBS and placed onto multi-well plates on which thawed MSCs at passage 4 had been seeded the previous day in complete medium at 10^4^ cells/cm^2^. Before placing the inserts, wells were washed with tempered PBS twice and media was changed to CTS-AIM-V medium supplemented with 0.1% Dipeptiven (Fresenius Kabi, Bad Homburg, Germany). The coculture (see Figure 1) was kept for 4 weeks under standard conditions and supernatants were collected over the course of a week. The different media supernatants were immediately frozen by immersion in dry ice and kept at −80 °C until analysis, at which point they were thawed at 4 °C and analyzed immediately after filtering through a 0.45 μM pore nitrocellulose filter (Merck KGaA, Darmstadt, Germany).

To obtain the secretome from MSC or from monocytes alone, the different cell populations were cultured separately under the same conditions as described for coculture. The mixture of the different cell culture media collections—once pooled to obtain the mixture of the 4-week process of coculture—constitutes the final product PRS CK STORM.

#### 2.1.3. Phenotypic Characterization of MSCs and M2 Macrophages by Flow Cytometry

For phenotypic characterization of MSCs and monocytes, samples were taken from both populations at times 0, 7, 14, and 28 days. MSCs were lifted with trypsin and monocytes with a scraper, and, after centrifugation at 300× *g* for 5 min at 4 °C, each cell type was resuspended in PBS. Monocytes were permeabilized with perm/wash buffer (BD Biosciences, Franklin Lakes, NJ, USA) and incubated for 30 min at 4 °C with the following fluorochrome-conjugated antibodies: CD68-FITC, CD163-PE, and CD206-APC (BD Biosciences, Franklin Lakes, NJ, USA).

MSCs were incubated identically with the following fluorochrome-conjugated antibodies: CD73-APC, CD45-FITC, CD31-PE, CD90-APC, and HLA-DR-FITC (Invitrogen, Waltham, MA, USA).

Fluorescence-minus-one technique was used to adjust voltages and compensate for fluorescence, and propidium iodide (Sigma, Burlington, MA, USA) was used to determine viability following the manufacturer’s instructions.

A Guava EasyCyte flow cytometer (Merck, Darmstadt, Germany) was used to acquire the samples, and InCyte software (Merck, Darmstadt, Germany) was used to analyze the results.

#### 2.1.4. Characterization of the Secretome (PRS CK STORM)

For the characterization of the secretomes of both cell types and coculture, 30 growth factors, cytokines, and chemokines were quantified using either ELISA or multiplex assay (Invitrogen, Grand Island, NY, USA) following the manufacturer’s instructions. A Luminex Labscan 100-plate reader (Luminex Corporation, Austin, Texas, USA) was used for determination. The molecules quantified by multiplex were the following: MIP1 alpha, IL-2, IL-6, TIMP-1, IL-8, IL-10, IL-12 P70, IL-1 RA, RANTES, GM-CSF, leptin, HGF, MMP-3, MCP1, BNGF, EGF, adiponectin, TNF alpha, MMP-1, TRAIL, FGF-2, PDGF-BB, and VEGF-A. For quantification of IGF-1, BMP-6, IL-1β, IL-4, TGF-β1, TGF-β3, and VEGF-C, commercial ELISA kits were employed (DuoSet ELISA kits, R&D, Minneapolis, MN, USA), and the manufacturer’s instructions were followed. Readings were performed on an iMark plate reader (BioRad, Hercules, CA, USA), and absorbances were measured at 450 and 570 nM.

#### 2.1.5. Anti-Inflammatory Potency Tests

The method adapted from Launay et al. 2013 [43] was used for this assay. PBMCs from healthy donors were seeded at a density of 1 × 10^6^ cells/mL in 100 µL of Gibco Roswell Park Memorial Institute (RPMI) 1640 + 10% FBS + 1% P/S in 96-well culture plates (Corning, Corning, NY, USA). PBMCs were stimulated with 100 pg/mL lipopolysaccharides (LPS) from Escherichia coli O111:B4 (Sigma–Aldrich, Burlington, MA, USA), and treated with 100 μL of PRS CK STORM or with the controls defined below in order to obtain a final volume of 200 μL. Hydrocortisone at 10 μg/mL (Sigma–Aldrich, Burlington, MA, USA) was used as anti-inflammatory control. Cells were treated/stimulated for 5 h, after which the supernatants were collected and frozen at −80 °C for cytokine analysis. In these supernatants variations the secretion of TNF-α and IL-1β, as main mediators of the response to lipopolysaccharide, were studied.

The THP-1 cell line was differentiated to macrophages by incubation with PMA and used also as an in vitro model to study biosafety and efficacy of the secretome. THP-1 cells (CellLineService, cat.nº.: 300356) were cultured and expanded in RPMI 1640 medium (Lonza, Basile, Switzerland) supplemented with 10% fetal bovine serum (FBS) (Corning, Corning, NY, USA), 1% penicillin/streptomycin (P/S) (Lonza, Basel, Switzerland) 1 mM sodium pyruvate (Lonza, Basel, Switzerland), and 1% MEM non-essential amino acids (Gibco-BRL, Waltham, MA, USA)—henceforth referred to as THP-1 medium. Cells were maintained at a density between 2 × 10^5^ and 10^6^ cells/mL to ensure adequate growth and a stable phenotype. At 48 h prior to LPS stimuli, cells were differentiated into resting macrophages using phorbol 12-mystyrate 13-acetate (PMA) (Sigma–Aldrich, Saint Louis, MO, USA) at 5 ng/mL in THP-1 medium as described in the protocol used by Park et al. [44]. After differentiation, cells were used for our experiments. All cell cultures were maintained at 37 °C, 5% CO_2_, and 98% RH. Differentiated THP-1 cells were washed three times with 0.2 mL of tempered THP-1 medium without PMA and allowed to rest for 30 min before LPS stimuli. After cell media change, cells were treated with 10 ng/mL LPS (Sigma–Aldrich, Burlington, MA, USA) in RPMI 1640 medium and treated with 100 μL of PRS CK STORM or the control as defined above by adding tempered THP-1 medium to obtain 200 µL of cell culture per well. After 5 h of stimulation, the supernatants were collected and frozen at −80 °C for cytokine analysis, in which variations in the secretion of TNF-α and IL-1β, as main mediators of the response to lipopolysaccharide, were studied. All experimental conditions were assayed in triplicate to ensure sufficient statistical robustness.

#### 2.1.6. MTT Assay

In this assay, adapted from Chen et al. 2016 [45], PBMCs from 2 different donors, were seeded at 10^6^ cells/mL in C-10 medium (RPMI 1640 without phenol red, +10% FBS (Corning, Corning, NY, USA), +1% P/S (Gibco-BRL), +0.1% β-mercaptoethanol (Gibco-BRL, Waltham, MA, USA), +0.1% sodium pyruvate (Gibco-BRL, Waltham, MA, USA), +0.1% sodium bicarbonate (Gibco-BRL, Waltham, MA, USA) and stimulated with 100 pg/mL of LPS. Cells were treated/stimulated in the same way as in the previous assay, with the same test groups and controls but the assay was kept for 96 h. After 96 h, 10 μL/well of an aqueous solution (5 mg/mL) of tetrazolium blue (Merk, Darmstadt, Germany) was added. The MTT/tetrazolium blue solution was incubated for 4 h at 37 °C, 5% CO_2_, after incubation the plates were centrifuged at 600× *g*, for 7 min to precipitate the cells and formazan crystals, and, after removal of the medium, the formazan crystals were solubilized with 200 μL/well of DMSO. The plates were incubated at 37 °C for 10 min and shaken at 250 rpm using a plate shaker (JP Selecta, Abrera, Catalonia, Spain). The results were obtained by measuring the absorbance of each well at 570 nM in an iMark plate reader (BioRad, Hercules, CA, USA).

To test in vitro biosafety, the same MTT assay described above was performed, but cells were not stimulated with LPS to assess if the product induced any inflammatory response.

#### 2.1.7. Statistics and Images

All statistics were calculated with data from five different patients performed in independent experiments in triplicate. ANOVA test was performed to determine statistically significant differences between the experimental groups studied using GraphPad Prism software version 8.4.0 for Mac OS X (GraphPad Software, San Diego, CA, USA) to perform the calculations. The level of statistical significance was set at *p* < 0.05.

The micrographs were taken with a Nikon camera (Nikon Corporation, Tokyo, Japan) attached to a Zeiss inverted microscope (Zeiss, Jena, Germany) and processed by Nikon’s Nis-elements software (NIS-Elements F Ver5.21.00 for 64bit edition, 06/2020, Konan, Minato-ku, Tokyo, Japan)

## 3. Results

### 3.1. Isolation of MSCs and Monocytes

The MSC isolation yield was approximately 1 × 10^5^ cells per mL of lipoaspirate, and incubation for 16 days under the conditions described in the previous section was necessary to bring the culture to pass 4 (Figure 2A). Regarding monocyte cultures, most of the adherent cells at the time of plating had typical monocyte/macrophage morphology (Figure 2B). Changes in morphology over time could also be observed in both MSCs and monocytes, possibly related to entry into the senescence phase (Figure 2C,D).

### 3.2. Phenotypic Characterization of MSCs and M2 Macrophages

The results of flow cytometric characterization of monocytes from three cocultures at the times studied are detailed in Table 1. The CD68 marker is used as a macrophage identifier. CD163 is present on monocytes and macrophages [46]. CD206 is mostly found on M2-polarized macrophages [47]. It can be concluded that the polarization towards the M2 phenotype is increasing with the time of coculture. Representative monocyte labeling after 14 days in coculture is shown in Figure 3C–E. Phenotypic characterization of MSCs was performed using the phenotypic labelling values established by consensus to define a homogeneous adipose tissue MSC population, consisting of, among others: CD90 > 90%; CD73 > 90%; CD31 < 2%; CD45 < 2%, and HLA-DR < 2% [48,49,50,51]. Representative MSC labeling after 28 days in coculture is shown in Figure 3A,B. This figure shows that the MSC phenotype does not change with time in culture. Figure 3C,D shows that the macrophage M2 phenotype does not change with time in culture.

### 3.3. Characterization of the Secretome

Characterization of the main components of the secretome was performed using the mutiplex methodology (Table 2).

To obtain the secretome pattern, those molecules that could be quantified as being within the detection limits of the method used in each case were studied, and significant differences were sought in comparison with the values taken as a control (conditioned medium of M2-like monocytes/macrophages). Those values that were significantly different in all samples studied (five patients in triplicate) were considered to form a specific and reproducible pattern of monocyte secretome modification by coculture with MSCs.

Coculture showed significant increases mainly in the anti-inflammatory cytokines TIMP-1 and IL-1Ra, in the regenerative growth factors HGF and IGF-1, in metalloproteinases MMP1 and MMP-3, and in dual cytokine IL-6. In addition, coculture showed significant decreases mainly in cytokines involved in inflammatory processes, such as MCP-1, PDGF-BB, and VEGF-A (Table 2 and Figure 4).

### 3.4. Pro-Inflammatory Cytokine Release Assays to Assess the Anti-Inflammatory Potency of the Secretome

The results (measurement of the release of the main pro-inflammatory cytokines IL-1β and TNF-α upon LPS stimulation of PBMCs from six donors or of the promonocytic THP-1 cells) are shown in Figure 5. In these assays, it can be observed that both cell types are sensitive to LPS and that the secretion of pro-inflammatory cytokines decreases when LPS-stimulated cultures were treated either with the control anti-inflammatory agent hydrocortisone (HC) or with the secretome under study.

Figure 5A,B show that the secretome under study is able to reverse the effects of LPS in all PBMC donors examined. The specific secretion of IL-1β and TNF-α induced by LPS was reduced by a mean of 60% by PRS CK STORM. The effect of PRS CK STORM was not significantly different from the control anti-inflammatory agent hydrocortisone. Individual results show that there was large variability between donors, but the mean responses follow the same pattern of inflammatory cytokine secretion, with the secretome being able to prevent their secretion in all donors.

In Figure 5C,D, the results obtained upon LPS stimulation of THP-1 cells are shown. THP-1 cells stimulated with LPS secreted both IL-1β and TNF-α, and their concentration abruptly decrease upon incubation with HC or with PRS CK STORM: IL-1β levels went back to basal values (100% inhibition), while specific TNF-α secretion was reduced by 75%.

### 3.5. Effect of the Secretome on Cell Viability and on LPS-Induced Growth

In order to test if the secretome was toxic by itself on PBMC or on THP-1 cells, we performed cell growth assays using the MTT method. As observed in the left bars of Figure 6A or Figure 6B, the secretome by itself did not have any effect on viable cell number on both cell types. In addition, we tested the effect of LPS on the growth of these cells and the effect of PRS CK STORM compared with HC on this LPS-stimulated growth. As shown in the right part of the graphics, LPS stimulated the growth of PBMC and of THP-1 cells around 40% in 96 h, and this growth effect was completely prevented by HC or by the different doses of PRS CK STORM.

## 4. Discussion

Cytokine storm is a health problem induced by several infectious agents, including SARS-CoV2; it is also present in inflammatory pathologies and associated with the use of CAR T cells in antitumoral treatment [39,40]. No standard solution is available to solve this problem—new anti-inflammatory drugs are needed. Our work was oriented in this sense, combining in a new biological treatment the known anti-inflammatory potency of M2 macrophages [18] and MSC cells [24].

Many researchers and authors report the influence of various factors on the immunomodulatory capacities of MSCs, such as the ratio of MSCs used in the coculture [52], the freezing and thawing cycles used [53], etc., but the number of culture passes stands out among all of them. In fact, two reports showed that cellular bioenergetics and growth rate were affected by an increase in the passage number in the culture [54,55]. Most authors agree that the immunomodulatory capacities of MSCs are greater when the number of passages is low, with several studies indicating that the number of passages between which MSC cells retain their immunomodulatory properties are between 3 and 7 [56]. Our results show that the ratios of MSCs to M2 macrophages used in the cocultures, as well as the culture pass of these MSCs, are appropriate to obtain a high immunomodulatory capacity.

Since the cocultures and their controls of MSCs and monocytes were incubated for 28 days, changes in morphology and/or number of adherent cells could be observed over time. For monocytes cultured alone, the typical maximum culture time is usually about two weeks, except when using cell lines or starting from stem cells [57], but these options could lead to oncogenicity issues. Our single monocyte cultures had a senescent appearance after the second week, much like the same monocytes cocultured with MSCs for 4 weeks. Interestingly, MSCs withstand many doublings and, therefore, a long time in culture without changing their appearance or phenotype (but under defined optimal culture conditions) [58]. The two cell types in coculture maintained their typical characteristics longer, demonstrating the existence of nourishing and cooperative behavior between them. In fact, coculture has been used as a nutritional technique for demanding cells, such as embryonic cells [59]. In any case, both MSCs and monocytes had a rounded and shriveled appearance in the last week of coculture, consistent with dead cells detaching from the culture surface.

Regarding the phenotypic characterization of the cocultured cells, it seems that the MSC population did not undergo changes throughout the coculture time, with a phenotype in agreement with that published by consensus by IFATS for MSCs from adipose tissue [48]. Phenotyping of monocytes shows the expression of CD68, a typical macrophage marker, and CD206, a typical marker of anti-inflammatory M2-polarized macrophages [47,60]. The purity of monocytes obtained after isolation by adhesion to plastic is around 70%, with approximately 30% of mainly contaminating lymphocytes attached to them [61]. In the usual monocyte culture differentiation protocols, 50 ng/mL of M-CSF are used (in addition to other supplements); we managed, thanks to the contribution of the MSC secretome [62], to add only 10 ng/mL of M-CSF to our monocytes and to remove this supplement before establishing the coculture, so that this protein of recombinant origin does not appear in the final formula, preventing possible antigenicity problems [63]. The transmembrane scavenger receptor CD163 is expressed exclusively in monocytes (low expression) and macrophages (high expression) [46]. On the other hand, it appears that CD163- and CD206-labelling may be less intense in fresh cells than in immunolabelling of fixed and permeabilized cells.

Of the 30 molecules analyzed, all those with a pro-inflammatory role are below the detection limits [64] except for IL-6 and IL-8, but these two cytokines have an ambiguous role in the immune response and are involved in auto-regenerative processes [64]. On the contrary, cytokines related to the resolution of inflammation and various growth factors related to regenerative mechanisms can be detected. Significant increases were observed after coculture, especially in IL-1ra (an IL-1β antagonist with anti-inflammatory potential), HGF (related to regenerative processes), and TIMP-1 (an inhibitor of metalloproteases involved in tissue regeneration). We did not find molecules such as BMP-6, which other authors have associated with the MSC secretome and regenerative properties [65]. This may be due to the use of a serum-free medium specific to monocyte culture [66]. Some cytokines which were present in the control culture media were either concentrated or decreased in the coculture, while levels of others were practically unchanged. This study gives us an idea of what kind of relationships are established for the target molecules between monocytes and MSCs in coculture, emulating the in vivo situation in a regenerative process.

Adiponectin production is typical of adipose cells, from which MSCs originate, and is traditionally described as a molecule regulating fat and sugar metabolism but has also been found to be involved in anti-inflammatory and regenerative processes [67]. Our group is currently using potency tests and various proof-of-concepts to assess the benefit of using coculture secretome versus monocytes and MSCs. On the other hand, it is possible that the greater complexity of the coculture secretome also implies a greater complexity of regulatory elements, so the full characterization of this secretome will require the study of other functional components, such as other cytokines and growth factors, lipids, microRNAs, circular RNAs, micro vesicles, exoxomes, and apoptotic bodies.

Our conditioned medium is derived from a coculture of healthy human cells and attempts to replicate the regenerative environment that results from the resolution of an inflammatory process. This drug has a very complex and variable composition as it depends on the donor cell to make the product. Because it is so variable, it requires quality-control methods in addition to the usual analytical techniques to ensure its efficacy and safety. The cytokine-release and MTT methods presented in this study are very useful in determining the possible in vivo effects that will be observed in humans, although they provide only a partial picture of the inflammatory response. These assays are quick and simple and provide sufficient information to determine which conditioned medium is safe and effective in vitro. Furthermore, being able to test the potency and potential toxicity of individual batches means that, in the future, when this drug needs to be industrially scaled up, this can be done through mixtures of different batches while maintaining anti-inflammatory potency and being able to establish more uniform compositions through such mixtures. Other products similar to this one successfully employs a like blending strategy to obtain a homogeneous product from very heterogeneous components [68]. Given the variability to LPS stimulation observed in the reactivity of PBMCs from each donor, using a standardized cell line, such as THP-1, would be of great advantage, allowing greater reproducibility [69,70].

The inflammatory mediators studied were selected because they are the first and foremost pro-inflammatory factors secreted when PBMCs are stimulated. IL-1β and TNF-α have a wide range of functions, from cell proliferation to induction of apoptosis [71]. They are also at the end of the signaling pathways used to generate inflammation in our model, as IL-1β is secreted when the inflammasome is engaged and caspase 1 converts inactive pro-IL-1β into bioactive and secretable IL-1β, while TNF-α is released from the cell membrane when the nuclear factor kappa-beta (NF-kβ) pathway is activated [72]. Although these two cytokines are not the only ones responsible for inflammation, they are able to provide good insight into the inflammatory response derived from LPS stimulation, as they are the main factors studied by most research groups performing this type of model [6,42,73].

It should be noted that, at least in the in vitro study performed, the anti-inflammatory potency of PRS is equivalent to hydrocortisone. Corticoids are the anti-inflammatory drugs more used in the clinic, but they also exhibit multiple undesired secondary effects, as has been recently reviewed, also in the COVID-19 context [74,75]. Future pre-clinical studies will explore if the physiologically balanced PRS CK STORM composition will avoid these secondary effects.

## 5. Conclusions

The results show a specific molecular pattern of interaction between the two cell types studied, obtained by the crosstalk established between the two cell types, giving a secretome (PRS CK STORM) with a clear anti-inflammatory, antifibrotic, and regenerative profile. Coculture of MSCs with monocytes results in a reproducible and specific pattern of modification of the secretome of the latter. The cytokine release method, by precisely quantifying the secretion of inflammatory factors, offers a valid in vitro test to assess anti-inflammatory potency, allowing us to evaluate possible toxicity and providing additional information on the mechanism of action of PRS CK STORM. Taken together, the results suggest that the secretome from the coculture of M2 macrophages with MSCs (PRS CK STORM) may become a safe and effective biological drug to prevent and treat the main health issue of cytokine storm.

## 6. Patents

This research work has resulted in the patent PCT/EP2020/059365 “Composition for tissue regeneration, method of production and uses thereof”.

## Figures and Tables

**Figure 1 biomolecules-12-00534-f001:**
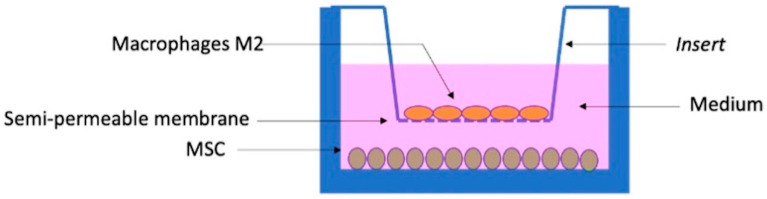
Schematic representation of a coculture in a 6-well plate. MSCs are adhered to the surface at the base of the well, while M2 macrophages adhere to the semi-permeable membrane of the insert, with no direct contact between the two cell types.

**Figure 2 biomolecules-12-00534-f002:**
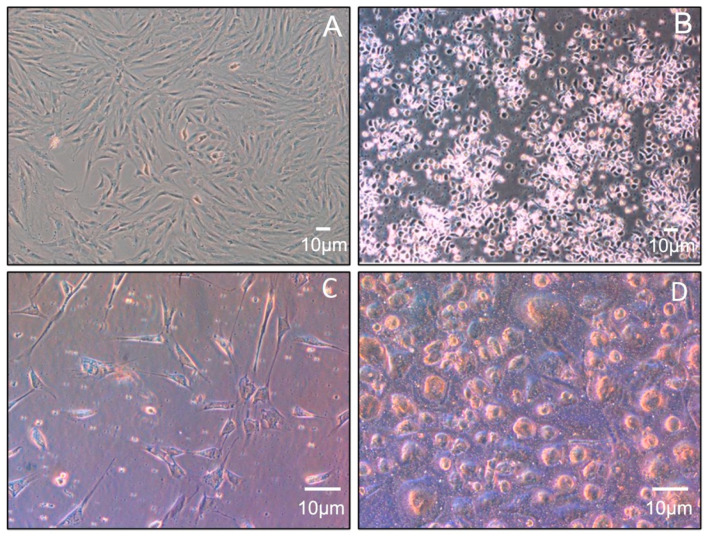
Representative micrographs (phase contrast) of cultures at different times. (**A**) Subconfluent MSCs at pass 3 (5×); (**B**) Adherent control cells at time 0 (5×); (**C**) MSCs at 28 days of coculture (20×); (**D**) M2-like monocytes/macrophages at 28 days of coculture (20×).

**Figure 3 biomolecules-12-00534-f003:**
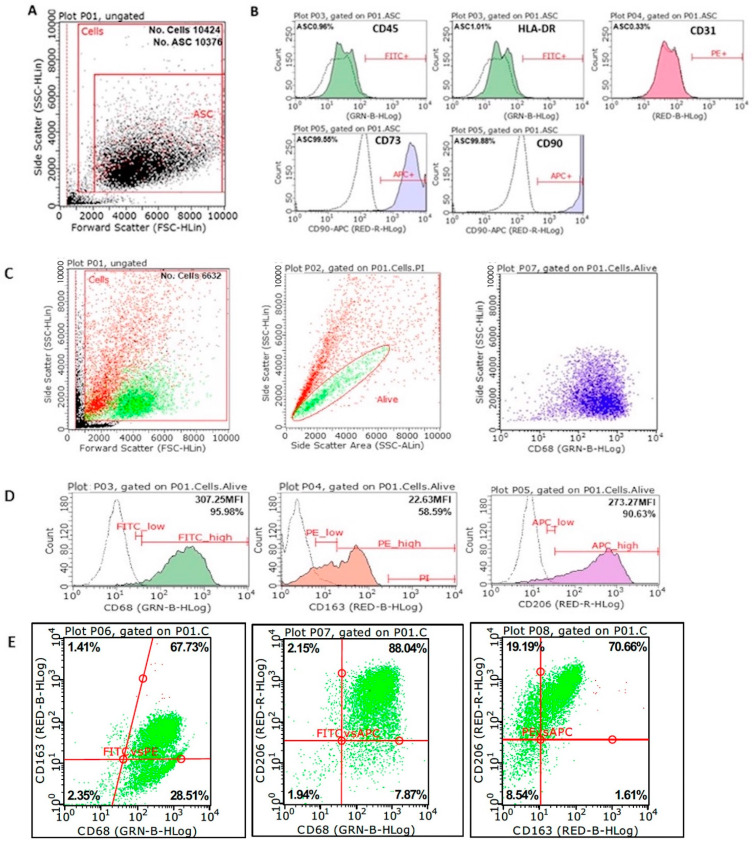
Representative flow cytometry results. Results are shown for MSCs at 28 days and monocytes at 14 days of coculture. (**A**) Acquired MSC population: the inner box selects living cells; (**B**) histograms of MSC labelling (CD45, HLA-DR, CD31, CD73, and CD90): the dotted line corresponds to the negative control or isotype. (**C**) Acquired monocyte population. Of all the events acquired, the smallest particles in FSvsSS corresponding to cellular debris have been eliminated. Next, with propidium iodide staining, it was verified that the FSlow cells belonged to dead cells (red) corresponded to the SS area low cells with a greater slope. Therefore, for the analysis, the cells included in the region of the SS area vs. SS SS-H graph (green) were selected. (**D**) Histograms of monocyte labelling (CD68, CD163, and CD206): the dotted line corresponds to the negative control or isotype. (**E**) CD163 vs CD68, CD206 vs CD68, and CD206 vs CD163 dotplots are shown.

**Figure 4 biomolecules-12-00534-f004:**
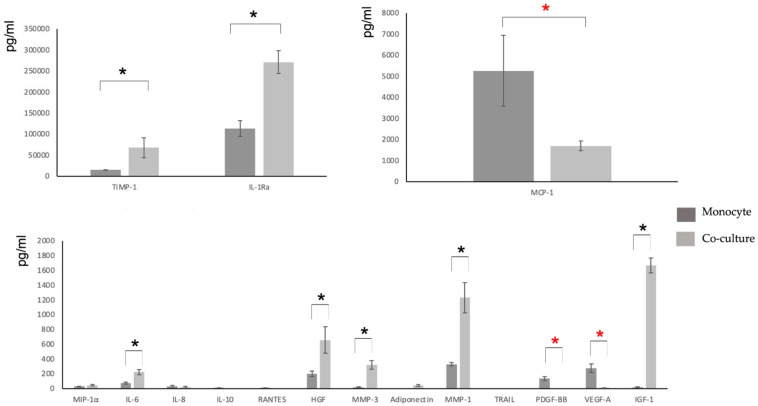
Cytokine secretion. Values of 16 selected soluble components are shown, comparing the monocyte with the coculture secretome. Stars mark significantly different values, black indicate an increase and red a decrease: *, *p*-values < 0.05.

**Figure 5 biomolecules-12-00534-f005:**
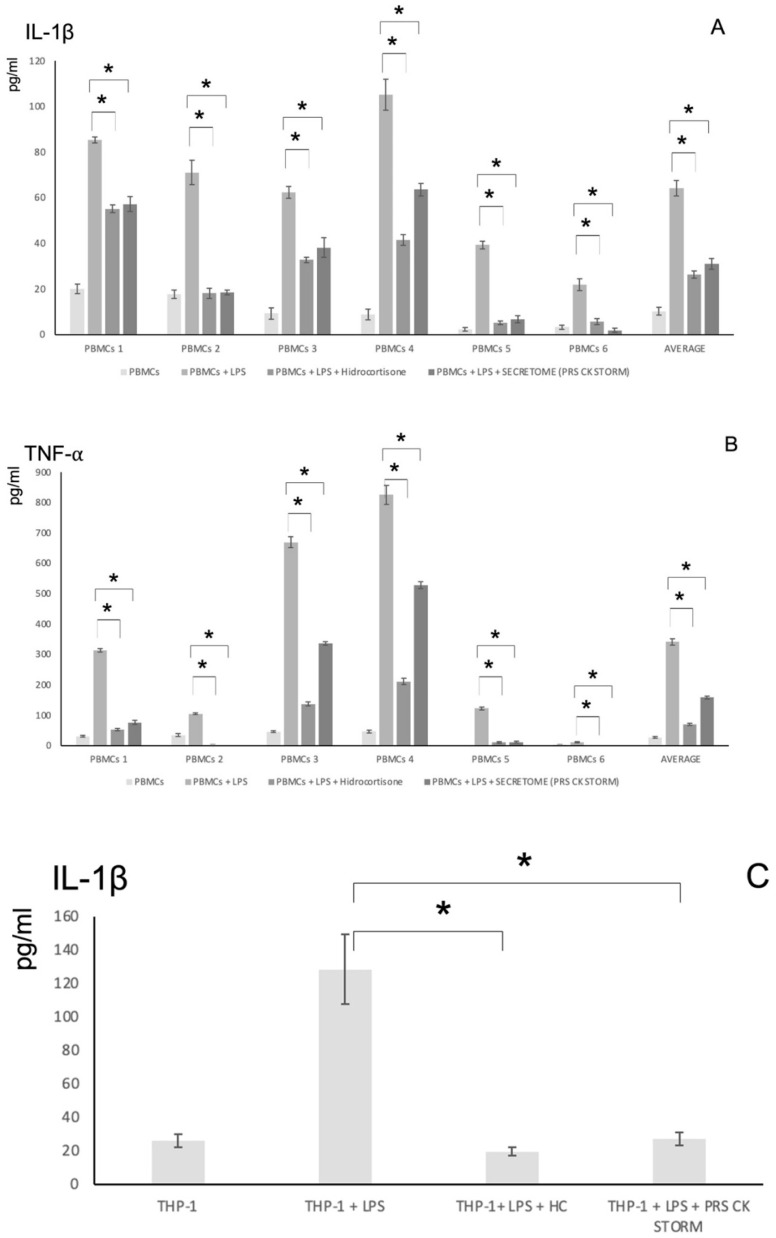
The release of IL-1β (**A**) or TNF-α (**B**) after treatment of PBMCs from 6 donors with 100 pg/mL of LPS for 5 h in the either the presence or absence (as indicated) of 10 μg/mL of hydrocortisone or the secretome was determined. The bars at the right of the graphics show the average results for the 6 donors. The release of IL-1β (**C**) or of TNF-α (**D**) after treatment of THP-1 cells with 10 ng/mL of LPS for 5 h in either the presence or absence (as indicated) of 10 μg/mL of hydrocortisone or the secretome was determined. Data show the mean values ± SD from three independent experiments with two replicates of the analytical technique in each case: *, *p*-values < 0.05.

**Figure 6 biomolecules-12-00534-f006:**
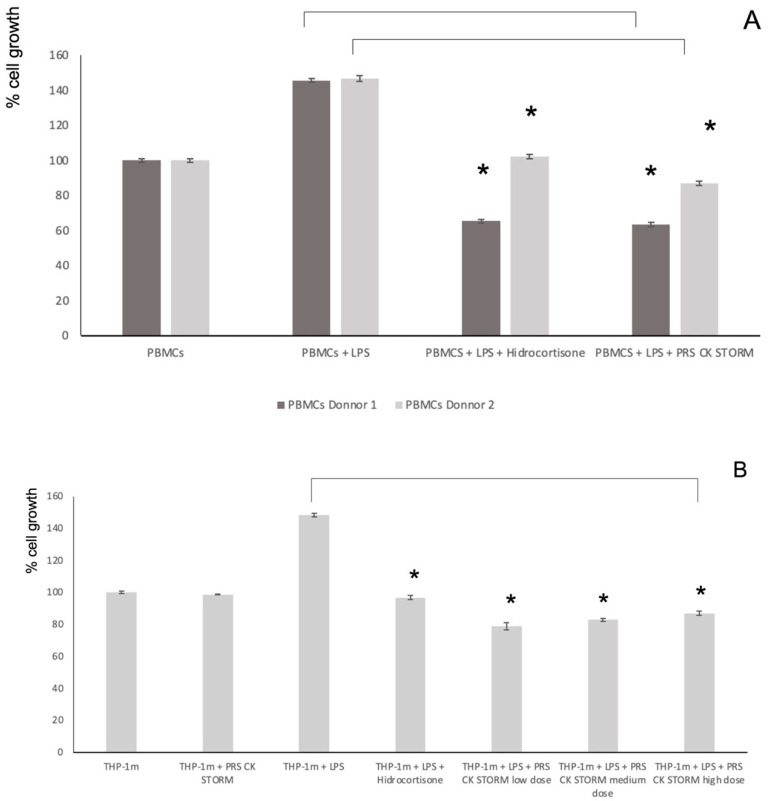
Estimation of cell growth using the MTT reduction method. Left bars: PBMCs from two different donors (**A**) or THP-1 cells (**B**) were cultured for 96 h in either the presence or absence (as indicated) of a high dose of PRS CK STORM, and MTT reduction was determined. Right bars: PBMC (**A**) or THP1 cells (**B**) were stimulated with 100 pg/mL or 10 ng/mL, respectively, for 96 h and cultured during this time in either the presence or absence (as indicated) of 10 µg/mL HC or with different doses of the secretome. The doses were low (×1), medium (×2.5), or high (×5). Data are expressed as percentage of cell growth compared with untreated control cells in each case and represent the mean ± SD of two experiments performed in triplicate: *, *p*-values < 0.05.

**Table 1 biomolecules-12-00534-t001:** Percentage of positive cells for each antibody tested in the monocyte culture. Immunolabelling performed fresh without fixation and permeabilization.

Sample	Co-Cultivation Time	CD68	CD163	CD206
**1**	7 days	94.48	42.04	73.21
	14 days	96.96	83.77	93.25
	28 days	82.82	71.21	82.33
**2**	7 days	93.51	73.25	64.2
	14 days	86.8	65.22	57.63
	28 days	75.9	50.17	76.77
**3**	7days	55.79	16.37	21.62
	14days	78.31	30.53	22.14
	28days	63.69	46.51	35.06

**Table 2 biomolecules-12-00534-t002:** Concentrations of the indicated soluble factors. Results are the mean ± SD of values obtained from 6 healthy monocyte donors. Values are shown in picograms per milliliter: <indicates that the value is below the detection limits; SD, standard deviation; NP, not applicable. The heatmap shows the same results in a color scale.

	**MIP-1α**	**IL-2**	**IL-6**	**TIMP-1**	**IL-8**	**IL-10**
**Monocyte control**	28.51(SD 5.05)	<7.21	75.4(SD 11.5)	14786.45(SD 989.56)	28.26(SD 15.71)	10.03(SD 1.67)
**MSC Control**	<2.234	<7.21	24.56(SD 9.68)	26789.32(SD 678.67)	18.77(SD 7.09)	0.06(SD 0.67)
**Co-culture MSC-M2**	46.07(SD 9.89)	<7.21	222.61(SD 35.8)	67788.33(SD 18776.9)	26.71(SD 9.91)	1.36(SD 1.12)
	**IL-12p70**	**IL-1Ra**	**RANTES**	**GM-CSF**	**IL-18**	**HGF**
**Monocyte control**	<4.71	113304(SD 23456.1)	10(SD 1.46)	<13.79	<12.21	199.23(SD 36.98)
**MSC Control**	1.08(SD 0.78)	<35	0.06(SD 0.09)	<13.79	<3.42	11.03(SD 2.81)
**Co-culture MSC-M2**	<4.71	271130(SD 27345.2)	2.92(SD 0.89)	<7.21	<12.21	654.78(SD 178.45)
	**MMP-3**	**MCP-1**	**BNGF**	**EGF**	**Adiponectin**	**TNF-α**
**Monocyte control**	18.9(SD 2.81)	5260(SD 1678.1)	<6.14	<1.78	<6.14	<1.45
**MSC Control**	94.42(SD 21.09)	193.88(SD 31.89)	1.84(SD 1.01)	<2.1	23.45(SD 4.88)	6.8(SD 1.81)
**Co-culture MSC-M2**	320.01(SD 56.67)	1687(SD 231.45)	<6.14	<1.78	44.16(SD 12.66)	3.19(SD 1.23)
	**MMP-1**	**TRAIL**	**FGF-2**	**PDGF-BB**	**VEGF-A**	**IGF-1**
**Monocyte control**	329.0 (SD 23.12)	1.13(SD 0.36)	6.53(SD 1.67)	135.33(SD 26.52)	275.58(SD 59.12)	19.32(SD 5.96)
**MSC Control**	180.26(SD 28.61)	1.23(SD 0.48)	0.44(SD 0.15)	3.12(SD 1.36)	241.24(SD 45.71)	434.5(SD 201.47)
**Co-culture MSC-M2**	1230.08(SD 204.96)	3.24(SD 1.2)	<2.72	3.95(SD 1.37)	9.82(SD 3.01)	1667.4(SD 102.78)
	**BMP-6**	**IL-1β**	**IL-4**	**TGF-β1**	**TGF-β3**	**VEGF-C**
**Monocyte control**	<6.14	<2.16	<10.49	< 10.49	<2.5	<12.21
**MSC Control**	<6.14	<2.16	<10.49	12.24(SD 3.39)	<2.5	<12.21
**Co-culture MSC-M2**	<6.14	<2.16	<10.49	450.85(SD 105.1)	<2.5	<12.21
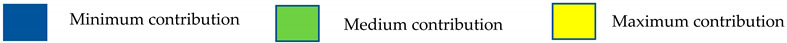

## Data Availability

The datasets generated and/or analyzed during the current study are available from the corresponding authors on reasonable request.

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
