# Peer review of "Cytokine Profile and Anti-Inflammatory Activity of a Standardized Conditioned Medium Obtained by Coculture of Monocytes and Mesenchymal Stromal Cells (PRS CK STORM)"

_biomolecules, 2022, doi:10.3390/biom12040534_

Round 1

Reviewer 1 Report

Manuscript ID: biomolecules-1638963

Title: Cytokine profile and anti-inflammatory activity of a standardized conditioned medium obtained by co-culture of monocytes and mesenchymal stromal cells (PRS© CK STORM)

To the authors:

Juan P Lapuente and colleagues described the composition of the secretome obtained from macrophages and MSCs co-culture, and demonstrated that the secretome presented similar anti-inflammatory effects to those observed for hydrocortisone, in vitro.

In my opinion, it is a very interesting work, very well conceived, and clearly described in the manuscript.

I suggest some minor revisions:

  1. “Introduction” section: please, correct “COVID-10” to “COVID-19”.
  2. Materials and Methods (“MSCs isolation” subsection): the authors indicated MSCs were cultured for the number of passages needed to obtain a homogeneous population of MSCs. Please, indicate (in the “MSCs isolation” subsection) the minimum and maximum number of passages used in this study. Several studies reported that the number of passages influence the immunomodulatory ability of MSCs. Have you compared the results obtained with MSCs from different passages? Have you found any differences? Please, add this information to the “Discussion” section and discuss the possible influence of using MSCs from different passages on the expression of cytokines and growth factors evaluated in this study.
  3. In the “PBMC and Monocyte isolation” subsection, the authors state they used MSCs in passage 4, which apparently contradicts the information given in the “MSCs isolation” subsection. Can you clarify this point, please?
  4. Figure 3: for the phenotypic characterization of monocytes (day 14) by flow cytometry, please, include the following dotplots in Figure 3: 1) SSC vs. CD68 and 2) CD163 vs. CD206.
  5. In the title of Table 2, the authors say “> and < indicate that the value is above or below the detection limits (respectively)”, but the “>” signal is never used in table 2, so it would be better to exclude it from the table 2 title.
  6. Looking at the results in Table 2, it is observed a decrease in IL-10 and an increase in TNFα levels in the MSC-M2 co-culture, compared to monocyte control. This is very interesting because it is unexpected, and I would like your input on this topic. Do you have any hypothesis to explain these results? Have you ever measured IL-10 and TNFα levels in a MSC-M2 co-culture in which cell contact was allowed, to investigate if membrane immunosuppressive molecules from MSCs could play a role in the regulation of IL-10 and TNF-α expression?
  7. Figure 5 would be more reader-friendly if you use horizontal bars in the graphs to indicate which groups have statistically significant differences. In Figure 5A and 5B it is not clear if the statistically significant difference was only detected for (PBMC + LPS) vs. (PBMC + LPS + PRS) or also for (PBMC + LPS) vs. (PBMC + LPS + Hydrocortisone). Also, the graphs will be easier to read if you add a title on the top of each graph, indicating which cytokine is being measured.

Author Response

  • 1ST REVIEWER

Title: Cytokine profile and anti-inflammatory activity of a standardized conditioned medium obtained by co-culture of monocytes and mesenchymal stromal cells (PRS© CK STORM)

To the authors:

Juan P Lapuente and colleagues described the composition of the secretome obtained from macrophages and MSCs co-culture, and demonstrated that the secretome presented similar anti-inflammatory effects to those observed for hydrocortisone, in vitro.

In my opinion, it is a very interesting work, very well conceived, and clearly described in the manuscript.

I suggest some minor revisions:

  1. “Introduction” section: please, correct “COVID-10” to “COVID-19”.

Thank you very much for the indication, the mistake has been corrected in the revised version.

  1. Materials and Methods (“MSCs isolation” subsection): the authors indicated MSCs were cultured for the number of passages needed to obtain a homogeneous population of MSCs. Please, indicate (in the “MSCs isolation” subsection) the minimum and maximum number of passages used in this study. Several studies reported that the number of passages influence the immunomodulatory ability of MSCs. Have you compared the results obtained with MSCs from different passages? Have you found any differences? Please, add this information to the “Discussion” section and discuss the possible influence of using MSCs from different passages on the expression of cytokines and growth factors evaluated in this study.

Thank you very much; it’s a very interesting appreciation; not only the culture passage number, if not another details like freeze and thawing cycles used, the ratio between MSC and macrophages, the medium, etc…Following your kind instructions we have included in the MSC isolation section a sentence and in the discussion section we have proceeded to open a debate on the subject, based on the current bibliography.

  1. In the “PBMC and Monocyte isolation” subsection, the authors state they used MSCs in passage 4, which apparently contradicts the information given in the “MSCs isolation” subsection. Can you clarify this point, please?

The reviewer is right, in the last edition of the previous version of the manuscript the sentence indicating that the cells were, once thawed, taken to pass 4 was lost. In the revised version the lost sentence has been added.

  1. Figure 3: for the phenotypic characterization of monocytes (day 14) by flow cytometry, please, include the following dotplots in Figure 3: 1) SSC vs. CD68 and 2) CD163 vs. CD206.

The suggestions of the reviewer are appreciated and the requested images have been added to Fig 3 in the revised version.

  1. In the title of Table 2, the authors say “> and < indicate that the value is above or below the detection limits (respectively)”, but the “>” signal is never used in table 2, so it would be better to exclude it from the table 2 title.

OK It is already corrected following your kind instructions

  1. Looking at the results in Table 2, it is observed a decrease in IL-10 and an increase in TNFα levels in the MSC-M2 co-culture, compared to monocyte control. This is very interesting because it is unexpected, and I would like your input on this topic. Do you have any hypothesis to explain these results? Have you ever measured IL-10 and TNFα levels in a MSC-M2 co-culture in which cell contact was allowed, to investigate if membrane immunosuppressive molecules from MSCs could play a role in the regulation of IL-10 and TNF-α expression?

This questions are really interesting and in my opinion, an issue to research in depth, but perhaps out of the scope of the present study. The contradictory result regarding the suppressive effect of MSC in culture without direct contact (1, 2) could be explained in part by the need of higher MSC: lymphocyte ratios (3). It appears that the ratio of MSCs in co-culture with direct contact positively influences the quantity and quality of direct intercellular contacts. Variable results have been obtained in studies analyzing contact and non-contact effects, mainly due to the effect of the microenvironment on induction, enhancement or maintenance of MSC immunoregulatory mechanisms. Although the importance of cell-cell contact for MSCs to effectively modulate the immune response has recently been highlighted, given that these interactions are difficult to achieve in a physiological context, it is quite possible that the release of extracellular vesicles (EVs) and their involvement as mediators of communication between MSCs and immune cells could be relevant.

Regarding the results obtained in the quantification of TNF-alpha and IL-10, it is true that they were somewhat unexpected. We are currently immersed in an investigation where we are trying to decipher the keys to the reason for this inversion of the expected values; however, we can affirm that this fact does not influence the anti-inflammatory immunomodulatory capacities of the conditioned medium obtained, given that we have already carried out in vivo tests in various species and the results have corroborated the good results obtained in vitro. We hope to be able to publish them shortly, although, I insist, we have to go much deeper into the mechanism of action of this conditioned medium, not only remaining in the merely cytokinic composition, but also deepening our proteomic knowledge of the medium. Thank you very much.

  1. Djouad F, Plence P, Bony C, Tropel P, Apparailly F, et al. (2003) Immunosuppressive effect of mesenchymal stem cells favors tumor growth in allogeneic animals. Blood 102: 3837-3844.

  1. Krampera M, Glennie S, Dyson J, Scott D, Laylor R, et al. (2003) Bone marrow mesenchymal stem cells inhibit the response of naive and memory antigen specific T cells to their cognate peptide. Blood 101: 3722-37229.

  1. Nasef A, Ashammakhi N, Fouillard L (2008) Immunomodulatory effect of mesenchymal stromal cells: possible mechanisms. Regen Med 3: 531-546.

  1. Figure 5 would be more reader-friendly if you use horizontal bars in the graphs to indicate which groups have statistically significant differences. In Figure 5A and 5B it is not clear if the statistically significant difference was only detected for (PBMC + LPS) vs. (PBMC + LPS + PRS) or also for (PBMC + LPS) vs. (PBMC + LPS + Hydrocortisone). Also, the graphs will be easier to read if you add a title on the top of each graph, indicating which cytokine is being measured.

Following the recommendation of the reviewer, the graphs have been adapted following your kind instructions in the revised version.

Thank you very much for the interest you have shown in reviewing our work. We deeply appreciate all the help provided.

Reviewer 2 Report

The manuscript by Juan Pedro Lapuente, et al. addresses an important issue about development of new biotherapy approach to treat cytokine storm, including those caused by COVID-19. It provides a detailed analysis of the secretome resulting from the co-cultivation of adipose MSCs and M2 macrophages. The study does not aim to deepen insights into inflammation mechanisms, but rather to analyze the composition and function of a putative biological product based on the conditioned medium. Nevertheless, this study may be of interest primarily to those involved in translational biomedicine. Increased content of the anti-inflammatory factors was identified by multiplex analysis in the co-culture medium in comparison with MSC or macrophage monocultures. The lack of cytotoxicity and ability to suppress both the proliferation of LPS-stimulated immune cells and pro-inflammatory IL-1b and TNF-alfa release were shown by MTT assay.  A disadvantage of the experimental design is the lack of data on the anti-inflammatory activity of secretomes from MSC and macrophage monocultures. It is well known that supernatants from both types of cultures have immunosuppressive and anti-inflammatory properties and MSC conditioned medium is even being tested in clinical trials to treat cytokine storm in COVID-19 patients (ClinicalTrials.gov Identifier: NCT04753476). Comparison of co-culture and monoculture within the study would be useful.

In general, the manuscript is well written, the conclusions are clear and supported by the data. The English language is good enough to understand the text clearly.  The number of self-citations is minimal. The methods described with sufficient details to allow another researcher to reproduce the results. While the experimental results are interesting the manuscript requires some improvement prior to publication:

  1. The studied culture medium is referred to as “PRS-CKSTORM”, “PRS©”, “PRS”, “PRS© CK STORM”, “Investigational Product (IP)”. The designation must be uniform.
  2. Table 1 caption. The assumption of a lower fluorescence intensity in live cells would be appropriate in the discussion section rather than in the caption.
  3. Table 1. Better to remove “macrophage” from first row of the table or add cell names to other CDs
  4. Figure 3a. A dot plot with FSS-H and FSS-A axis is not a common way to gate living cells. Besides, in the Methods section, it was mentioned that the macrophages were permeabilized. Explanations needed.  
  5. Figure 3d. The pick is out of the axis range.
  6. Table 2. “>” is mentioned in the caption but not used in the table.
  7. Figure 5 a,b. There is a large variation in values among different donors. While the averages have low variance as seen from the whiskers on their bars. Explanations needed.
  8. Figure 6 b. How to explain the lack of a dose-dependent effect.
  9. Discussion. The reference [64] refers to the work of the authors, which demonstrates that in the presence of human serum, MSCs do not undergo spontaneous transformation in vitro. This contradicts what is stated in the manuscript.

Author Response

  • 2ST REVIEWER:

The manuscript by Juan Pedro Lapuente, et al. addresses an important issue about development of new biotherapy approach to treat cytokine storm, including those caused by COVID-19. It provides a detailed analysis of the secretome resulting from the co-cultivation of adipose MSCs and M2 macrophages. The study does not aim to deepen insights into inflammation mechanisms, but rather to analyze the composition and function of a putative biological product based on the conditioned medium. Nevertheless, this study may be of interest primarily to those involved in translational biomedicine. Increased content of the anti-inflammatory factors was identified by multiplex analysis in the co-culture medium in comparison with MSC or macrophage monocultures. The lack of cytotoxicity and ability to suppress both the proliferation of LPS-stimulated immune cells and pro-inflammatory IL-1b and TNF-alfa release were shown by MTT assay.  A disadvantage of the experimental design is the lack of data on the anti-inflammatory activity of secretomes from MSC and macrophage monocultures. It is well known that supernatants from both types of cultures have immunosuppressive and anti-inflammatory properties and MSC conditioned medium is even being tested in clinical trials to treat cytokine storm in COVID-19 patients (ClinicalTrials.gov Identifier: NCT04753476). Comparison of co-culture and monoculture within the study would be useful.

In general, the manuscript is well written, the conclusions are clear and supported by the data. The English language is good enough to understand the text clearly.  The number of self-citations is minimal. The methods described with sufficient details to allow another researcher to reproduce the results. While the experimental results are interesting the manuscript requires some improvement prior to publication:

  1. The studied culture medium is referred to as “PRS-CKSTORM”, “PRS©”, “PRS”, “PRS© CK STORM”, “Investigational Product (IP)”. The designation must be uniform.

Following the recommendation of the reviewer, we have termed our secretome as PRS CK STORM throughout the entire manuscript.

  1. Table 1 caption. The assumption of a lower fluorescence intensity in live cells would be appropriate in the discussion section rather than in the caption.

We agree with the reviewer, in the revised version the text has been changed.

  1. Table 1. Better to remove “macrophage” from first row of the table or add cell names to other CDs

This has been corrected in the revised version following your kind instructions.

  1. Figure 3a. A dot plot with FSS-H and FSS-A axis is not a common way to gate living cells. Besides, in the Methods section, it was mentioned that the macrophages were permeabilized. Explanations needed.  

Many thanks to the reviewer, there is an error in the explanation of figure 3 that had gone unnoticed by the authors. Figure 3A corresponds to the population of monocytes analyzed, in which the living cells have been marked thanks to the identification, by means of propidium iodide (PI), of the dead cells in a previous dot plot. Monocyte viability was analyzed in a separate, unfixed aliquot by PI. This is explained in the revised version.

  1. Table 2. “>” is mentioned in the caption but not used in the table.

This mistake has been corrected following the instructions of the reviewer.

  1. Figure 5 a,b. There is a large variation in values among different donors. While the averages have low variance as seen from the whiskers on their bars. Explanations needed.

We attach a table where you can see that the results of the SD of the mean are calculated correctly

  1. Figure 6 b. How to explain the lack of a dose-dependent effect.

Although perhaps a great difference is not observed in the graph in the three doses used, it is true that when the dose is increased a slight improvement in the response is seen (A table is shown to be able to assess the results); it is true that it does not follow a typical dose-dependent response. We believe that this fact may be due to the fact that the three doses used are very close to the MABEL dose (Minimum Anticipated Biological Effect Level)

  1. Discussion. The reference [64] refers to the work of the authors, which demonstrates that in the presence of human serum, MSCs do not undergo spontaneous transformation in vitro. This contradicts what is stated in the manuscript.

The reviewer is right, the text was incorrect in the previous version of the manuscript.  The authors should have explained that the elevation of the c-myc protein, an oncogene, could be suspicious, at least from a regulatory point of view, for its future clinical use. However, the idea is based on conjecture, so the reference and the phrase have been removed in the revised version.

Thank you very much for the interest you have shown in reviewing our work. We deeply appreciate all the help provided.

Reviewer 3 Report

In the article entitled "Cytokine profile and anti-inflammatory activity of a standardized conditioned medium obtained by co-culture of monocytes
and mesenchymal stromal cells (PRS© CK STORM)", the authors tried to show the anti-inflammatory role of conditioned medium obtained by co-culture of monocytes and mesenchymal stromal cells. Few points to be noted:

  1. The authors should include a scale for the microscopic images in Fig2.
  2. The authors should improve the Flowcytometry data in Fig3. The population should be represented in a dot plot that clearly distinguishes the population.  
  3. For Table 2: the authors can choose Bar Diagram for each cytokine or a heat map representation can be done. It is not very clear in the table and it is very messy with all the numbers scattered and not readable.
  4. Fig5: Y-axis should be properly arranged.  
  5. Please check the grammatical errors. 

Author Response

  • 3ST REVIEWER

In the article entitled "Cytokine profile and anti-inflammatory activity of a standardized conditioned medium obtained by co-culture of monocytes
and mesenchymal stromal cells (PRS© CK STORM)", the authors tried to show the anti-inflammatory role of conditioned medium obtained by co-culture of monocytes and mesenchymal stromal cells. Few points to be noted:

  1. The authors should include a scale for the microscopic images in Fig2.

Following the kind instructions of the reviewer, we added to the image a scale bar

  1. The authors should improve the Flowcytometry data in Fig3. The population should be represented in a dot plot that clearly distinguishes the population.  

Following the suggestion of the reviewer, and also from another referee, we have modified Figure 3 in the revised version, including more flow cytometry information. 

  1. For Table 2: the authors can choose Bar Diagram for each cytokine or a heat map representation can be done. It is not very clear in the table and it is very messy with all the numbers scattered and not readable.

Following your kind instructions, we have changed table 2 to table 2 which includes also a heatmap

  1. Fig5: Y-axis should be properly arranged.  

Following your kind instructions, we have changed the Y axis in all the graphs

  1. Please check the grammatical errors. 

Thank you very much; All the document has been checked

Thank you very much for the interest you have shown in reviewing our work. We deeply appreciate all the help provided.

Round 2

Reviewer 1 Report

The authors addressed all comments and answered all questions raised in the review process.

Author Response

Thank you very much for the interest you have shown in reviewing our work. We deeply appreciate all the help provided

Reviewer 2 Report

In its second version, the manuscript has been improved, but the text still contains inaccuracies that need to be corrected.

  1. Row 287. In the updated manuscript the Fig 3 A, B represents data on MSC but not monocyte staining. Row 291. Fig C, D – monocytes but not MSC.
  2. Fig 3B upper line. X-axis is labeled as “CFSE” or “PI”. Even if it was automatically generated as a fluorescence channel name, it should be removed or changed to the correct one. Otherwise, it may confuse a reader.
  3. Fig 3E not mentioned in the figure caption.
  4. Table 2. It is generally not recommended to present the same data in different ways. You can use a supplement if needed.
  5. Row 84. It's better to remove the text within the brackets because "zoonoses or para-sitic and/or fungal infections" are not infectious agents. Moreover, the reference [33] refers only to viral infection.
  6. Rows 127, 209. The abbreviation was used before the full name (222, 221).
  7. Rows 209, 242. There is probably a typo. FBS, not SBF.

Author Response

(The authors gave the same response as above.)

Reviewer 3 Report

The authors accepted all the suggestions and modified the manuscript accordingly.